

# Anywhere but here: local conditions motivate dispersal in *Daphnia*

Philip Erm[1], Matthew D. Hall[2] and Ben L. Phillips[1]

[1] School of BioSciences, University of Melbourne, Parkville, VIC, Australia
[2] School of Biological Sciences, Monash University, Clayton, VIC, Australia

## ABSTRACT

Dispersal is fundamental to population dynamics. However, it is increasingly apparent that, despite most models treating dispersal as a constant, many organisms make dispersal decisions based upon information gathered from the environment. Ideally, organisms would make fully informed decisions, with knowledge of both intra-patch conditions (conditions in their current location) and extra-patch conditions (conditions in alternative locations). Acquiring information is energetically costly, however, and extra-patch information will typically be costlier to obtain than intra-patch information. As a consequence, theory suggests that organisms will often make partially informed dispersal decisions, utilising intra-patch information only. We test this proposition in an experimental two-patch system using populations of the aquatic crustacean, *Daphnia carinata*. We manipulated conditions (food availability) in the population's home patch, and in its alternative patch. We found that *D. carinata* made use of intra-patch information (resource availability in the home patch induced a 10-fold increase in dispersal probability) but either ignored or were incapable of using of extra-patch information (resource availability in the alternative patch did not affect dispersal probability). We also observed a small apparent increase in dispersal in replicates with higher population densities, but this effect was smaller than the effect of resource constraint, and not found to be significant. Our work highlights the considerable influence that information can have on dispersal probability, but also that dispersal decisions will often be made in only a partially informed manner. The magnitude of the response we observed also adds to the growing chorus that condition-dependence may be a significant driver of variation in dispersal.

# INTRODUCTION

Dispersal, like survival and reproduction, is a fundamental facet of life history (*Bonte & Dahirel, 2017*). Behaviours which govern dispersal can have profound consequences for a variety of biological and ecological phenomena, such as individual fitness, metapopulation dynamics, and evolutionary outcomes across a species' range (*Bonte & Dahirel, 2017*; *Clobert et al., 2001*). For reasons of simplicity, the majority of spatially-explicit ecological models assume that dispersal is both uninformed and unchanging; that individuals disperse at fixed rates, and that they do so without recourse to information about environmental

Corresponding author
Philip Erm,
philip.erm@unimelb.edu.au

conditions (e.g. by default in models utilising reaction-diffusion or integrodifference equations; *Fisher, 1937*; *Skellam, 1951*; *Kot, Lewis & Van Den Driessche, 1996*; but see *Fronhofer, Nitsche & Altermatt, 2016*). There is now considerable evidence, however, that dispersal decisions are routinely informed by aspects of the environment (*Clobert et al., 2009*), with such information use expected to have non-negligible effects on ensuing population and evolutionary dynamics (*Delgado et al., 2014*; *Ponchon et al., 2015*; *Urban et al., 2016*).

The most common form of informed (or condition-dependent) dispersal is density-dependent dispersal (*Bowler & Benton, 2005*). Here, individuals acquire information about population density, and, if conditions more favourable to survival and reproduction are likely to be found elsewhere, make the decision to disperse. When all else is equal, high density—with its greater competition for resources, greater rates of disease transmission and so on—will be associated with poorer conditions (*Bowler & Benton, 2005*). Many species have been shown to acquire information on density and act upon it, such as the spider *Erigone atra* (*De Meester & Bonte, 2010*), protozoa like *Paramacium caudatum* and *Tetrahymena thermophila* (*Fellous et al., 2012*; *Fronhofer, Gut & Altermatt, 2017*), and the plant *Heterosperma pinnatum* (*Martorell & Martínez-López, 2014*). In many species, this information is acquired through food availability; when local resources are limited, individuals tend to be more dispersive, regardless of taxa. This was demonstrated by *Fronhofer et al. (2018)*, where resource limitation (and to a lesser extent, predator presence) was shown to induce higher dispersal rates in organisms as varied as, amongst others, protists, slugs, crustaceans, crickets, newts, and fish. Information on the relative merits of different alternative locations can also be acquired in numerous ways, including prospecting (e.g. actively assessing potential breeding sites, as in collared flycatchers (*Ficedula albicollis*) monitoring which locations are producing well-provisioned nests; *Pärt & Doligez, 2003*) and by observing immigrating conspecifics (as in the common lizard *Zootoca vivipara*; *Cote & Clobert, 2007*).

Several models have now been constructed to examine the evolution of informed dispersal (*Bocedi, Heinonen & Travis, 2012*; *Delgado et al., 2014*). However, while they are in general agreement that informed dispersal will often evolve, much hinges on the ease with which information is acquired, along with its value (*Poethke et al., 2016*). In an ideal world, we would expect dispersal decisions to be made on a balance of 'push' factors, such as local patch conditions, and 'pull' factors, such as the quality of other patches. The quality of the new patch should be high enough (relative to the home patch) that it offsets the fitness costs of moving. But organisms do not live in an ideal world: information can be costly to acquire, both in terms of time and energy (*Bonte et al., 2012*), with a likely asymmetry of cost such that information about alternative patches is harder to obtain than information about an individual's home patch. Thus, it may often be the case that individuals act on the limited information that is most easily acquired: intra-patch information.

Conversely, ecological and population dynamics may also be influenced by the relative strength of the pull exerted by extra-patch information. In metapopulations with dispersers that can take advantage of this kind of information, patch persistence may be

expected to increase if patches with perilously low population sizes—but abundant resources as a result—become more appealing to dispersers. Indeed, simply being able to detect other conspecifics in these low-density patches may have the same effect (*Clobert et al., 2009*), although patches with suitable habitat could also become overpopulated if they attract a disproportionate number of migrants. In biological invasions, invasion speed may be boosted if colonisers are able to use extra-patch information to select suitable habitats, or hindered if it instead causes them to favour ecological traps (*Kokko, 2006*). If intra-patch information is dominant in motivating dispersal on the other hand, invaders may be expected to distribute themselves indiscriminately, rendering the invasion highly sensitive to both the proportion of suitable habitat in the landscape and any temporal fluctuations in its quality (*Neubert, Kot & Lewis, 2000*; *Schreiber & Lloyd-Smith, 2009*). Gauging the relative strength of the push caused by intra-patch information and the pull caused by extra-patch information may help to resolve such questions.

Here, we examine the relative influence of intra- and extra-patch information on dispersal by manipulating food resource levels in experimental populations of the aquatic crustacean, *Daphnia carinata*. Dispersal in *Daphnia* is usually characterised as being driven by the passive transport of ephippia (long-lived resting eggs) by water fowl or other vectors (*Allen, 2007*; *Frisch, Green & Figuerola, 2007*; *Van De Meutter, Stoks & De Meester, 2008*); however, individuals can also actively disperse between permanently or temporarily interconnected water bodies (*Michels et al., 2001*; *Cottenie et al., 2003*). Although it has been demonstrated that *Daphnia* do boost ephippia production in response to information cues indicating low local resource availability (*Carvalho & Hughes, 1983*; *Hobaek & Larsson, 1990*; *Kleiven, Larsson & Hobaek, 1992*), a greater range of responses has been observed regarding its effects on active movement. Environments with relatively higher food concentrations have been shown to increase *Daphnia* movement behaviours like swimming speed and sinking rate (*Dodson et al., 1997*); however, in other instances, they have been shown to slow movement, with much depending on the *Daphnia* species or clone line under examination (*Young & Getty, 1987*; *Larsson & Kleiven, 1996*; *Roozen & Lürling, 2001*). *Daphnia* have also been seen to adhere to ideal free distributions under ordinary circumstances, with individuals favouring regions of high food concentration so long as they fall within natural ranges (*Jakobsen & Johnsen, 1987*; *Neary, Cash & McCauley, 1994*; *Jensen, Larsson & Högstedt, 2001*). It would appear likely then, that *Daphnia* exploit information to regulate their dispersal efforts between patches. It is less clear however, if this behaviour is governed entirely by intra-patch information, or if extra-patch information also influences dispersal propensity.

Using *D. carinata*, we determine if individuals modify their rates of active dispersal between patches in small multi-patch mesocosms when exposed to different intra-patch resource levels. We also ask whether this response is contingent upon extra-patch conditions; the presence or absence of ad libitum food in the neighbouring patch.

## MATERIALS AND METHODS

### Laboratory population of *D. carinata*

All *D. carinata* used were genetically identical members of a single clone line. The founding member of this lineage was collected at 3810′34.3″S, 14421′14.1″E (a lake in Geelong, VIC, Australia) in October 2016. Its offspring were used to establish laboratory stock cultures, which were housed in glass jars containing 300 ml of ADaM zooplankton medium (according to the recipe of *Klüttgen et al., 1994*; as modified by *Ebert, Zschokke-Rohringer & Carius, 1998* using only 5% of the recommended $SeO_2$ concentration) and kept within growth chambers maintained at 22 °C on a 12.30 light:11.30 dark photoperiod. Stocks were fed the non-motile green algae *Scenedesmus*. In order to reduce any potential impact of maternal effects, all individuals used in the experiment were taken from stocks that were maintained under these conditions for at least two generations.

### Experimental materials and conditions

We set up two-patch microcosms within which to measure dispersal of *Daphnia*. Each patch was a 950 ml plastic Cryovac XOP-0980 container filled with 600 ml of ADaM and kept on bench tops in an open-air laboratory. The laboratory was maintained at 22 °C and each container was covered with a transparent plastic sheet that was only removed during feeding and data collection. Each container measured 90 × 75 mm at the base, was 110 mm high and widened gradually towards the top to 100 × 90 mm. A circular hole with a diameter of 15 mm was centrally located 35 mm above the base on one of the long sides of each container. This was connected to plastic PVC piping of an identical internal diameter that linked one container to the next, acting as a 117 mm long tunnel through which *D. carinata* could disperse between the two containers. For *Daphnia* generally, such a length would be easily traversable within less than a minute for an individual swimming in a straight line (*O'Keefe, Brewer & Dodson, 1998*). At the commencement of the experimental trials, dispersal between containers was prevented by inserting cotton balls into the openings of the connecting tunnel.

### Food availability experiment

Within this two-patch system, we examined the effects of intra- and extra-patch food availability on the dispersal rate of *D. carinata*. We seeded one half of each two-patch system with 10 adult females taken from stock cultures, and allowed this population to grow for 9 days in the experimental system while dispersal was blocked. This resulted in each population containing individuals of a variety of age and size classes when dispersal commenced (*D. carinata* have a lifespan of 1–2 months depending on the conditions at which they are maintained, and generally reach reproductive maturity when between 5 and 10 days old; *Venkataraman & Job, 1980*). On the 10th day, we then unblocked the dispersal tunnels and made one exhaustive count by eye of the number of adult (i.e. individuals large enough to reproduce) and juvenile *D. carinata* in each patch every 24 h thereafter for 4 days.

Our patch pairs were allocated to four treatment combinations ($n = 5$ per combination) according to a two factor crossed design in which we independently modified food

availability in the two patches. Factor 1 was intra-patch food availability: once the dispersal tunnel was unblocked, half of the populations no longer received food in their starting patch. Factor 2 was extra-patch food availability: here we either daily added food to the second patch (commencing on day 7, 3 days before dispersal was allowed) or withheld food altogether from this patch. This meant that half of the populations were dispersing into patches that contained no food at all, and the other half into patches with an abundance of food. Food in this case was a daily fed mixture of 8 million *Scenedesmus* sp. cells (an unidentified Australian *Scenedesmus*) and 12 million *Scenedesmus obliquus* cells.

We examined the effect of feeding regimes on absolute population sizes using ANOVA. There were two response variables: the total population size at 96 h (summed across both patches); and the population size in patch 2 at 96 h. We verified that they did not violate standard ANOVA assumptions by testing each for normality and homogeneity of variances using the Shapiro–Wilk test and Levene's test. Neither assumption was violated for population size at 96 h (Shapiro–Wilk: $W = 0.92659$, $P = 0.133$; Levene's: $F = 0.910$, $P = 0.458$) nor population size in patch 2 at 96 h (Shapiro–Wilk: $W = 0.93031$, $P = 0.157$; Levene's: $F = 1.620$, $P = 0.224$). We next compared the proportion of individuals that had reached patch 2 after 96 h between treatment combinations using a generalised linear model with binomial errors and a logit link, with each individual in each patch being characterised as a trial in which either success (dispersing into patch 2) or failure (remaining in patch 1) had resulted. We likewise used a generalised linear model with binomial errors and a logit link to see if adults or juveniles were over-represented as dispersers after 96 h. Finally, we also examined the relationship between the proportion of individuals dispersing and density using a generalised linear model with quasibinomial errors and a logit link. A quasibinomial error distribution was selected because there was a large difference between the residual deviance of the binomial model (61.287) and its degrees of freedom (16).

All statistical tests were performed in R version 3.5.0 (*R Development Core Team, 2018*). All experimental data and scripts are available in the figshare repository at https://doi.org/10.4225/49/5b0f62dc23b4c.

## RESULTS

Over the course of the dispersal phase, total population sizes across both patches generally increased or decreased according to whether patch 1 was fed or not, with fed treatments overall growing in size and unfed treatments shrinking (Fig. 1). At 96 h, we found a significant effect of food availability in patch 1 on total population size across both patches ($F_{1,16} = 10.826$, $P < 0.01$; Table 1), but not of food availability in patch 2 ($F_{1,16} = 0.013$, $P = 0.912$; Table 1). The interaction between feeding treatments in the two patches was also not significant with regard to total population size ($F_{1,16} = 0.481$, $P = 0.498$; Table 1).

Examining the proportion of individuals dispersing after 96 h, we found no significant effect of the interaction between intra- and extra-patch feeding treatments ($z = 1.073$, $P = 0.283$; Table 2), and likewise no significant effect of food availability in patch 2 ($z = 0.138$, $P = 0.890$; Table 2). We did however find a significant effect of food availability
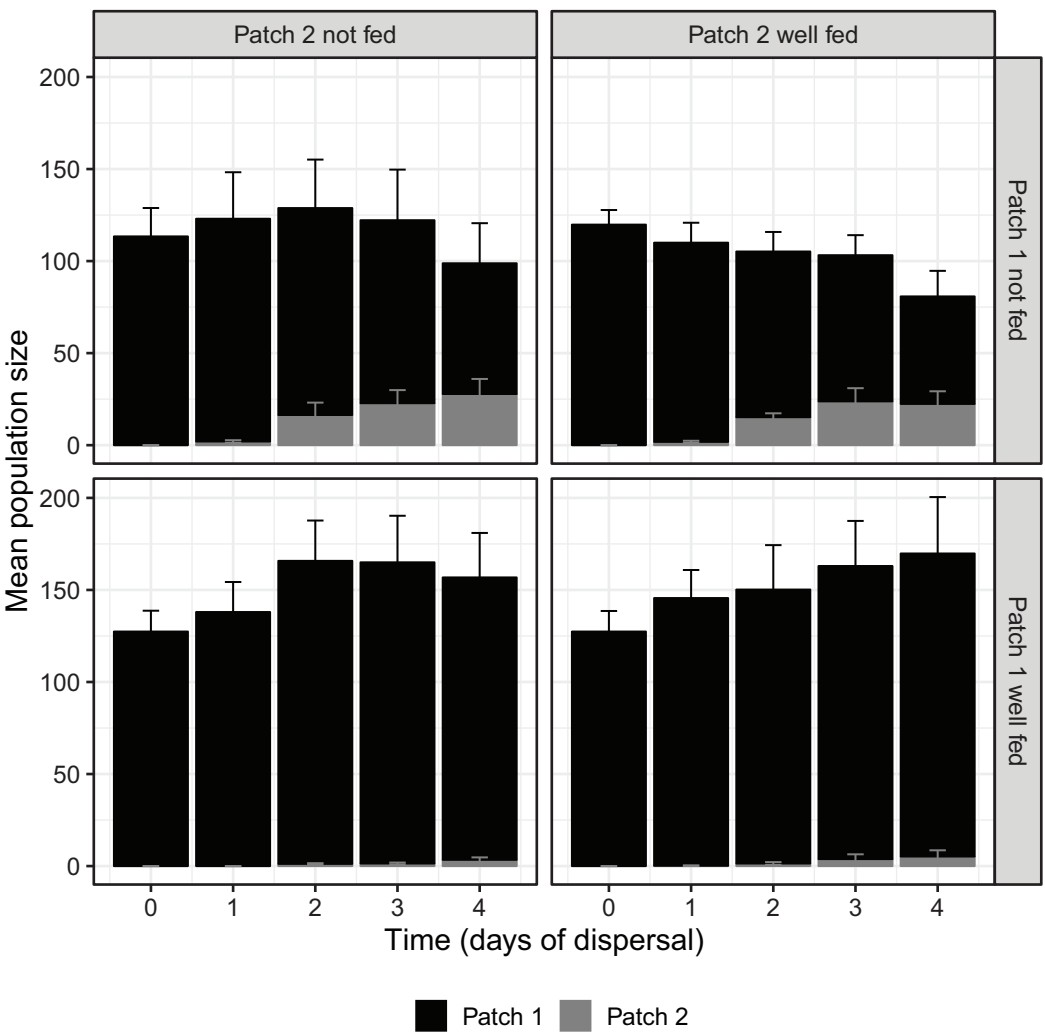

**Figure 1** *D. carinata* **population size over time and space across distinct feeding regimes.** The effect of feeding regime on mean population size across both patch 1 and patch 2 over the dispersal phase (*n* = 5 container pairs per treatment combination). Bars are stacked, such that both patch 1 (black) and patch 2 (grey) population sizes combine to indicate mean population size. Error bars show SE for each patch's mean population size, rather than for stacked mean population size. Replicates where patch 1 was not fed produced higher numbers of dispersers despite having lower total population sizes (Table 1).

in patch 1 ($z = 10.843$, $P < 0.001$; Table 2), with intra-patch food deprivation resulting in an approximately 10-fold higher proportion of the total population dispersing (food-deprived patch 1, mean = 0.259, SE = 0.0374; well-fed patch 1, mean = 0.0218, SE = 0.00718; Fig. 2).

Since total population size did differ based on food availability in patch 1, we examined its relationship with the proportion of *D. carinata* dispersing. Although the proportion of dispersers in both the patch 1 fed and unfed groups appeared to increase with higher densities (food-deprived patch 1, $r^2 = 0.234$; well-fed patch 1, $r^2 = 0.401$; Fig. A1), we did not find a significant effect of density on the proportion of dispersers ($t = 1.273$, $P = 0.221$; Table A1). We likewise found no significant effect of the interaction

**Table 1 Statistical results for total population sizes and absolute number of dispersers after 96 h of dispersal.**

| Source | df | F stat | P-value |
| --- | --- | --- | --- |
| Total population sizes | | | |
| Food available in patch 1 | 1 | 10.826 | <0.01 |
| Food available in patch 2 | 1 | 0.013 | 0.912 |
| Food available in patch 1 × Food available in patch 2 | 1 | 0.481 | 0.498 |
| | 16 | | |
| Absolute number of dispersers | | | |
| Food available in patch 1 | 1 | 13.605 | <0.01 |
| Food available in patch 2 | 1 | 0.102 | 0.754 |
| Food available in patch 1 × Food available in patch 2 | 1 | 0.408 | 0.532 |
| | 16 | | |

Note:
ANOVA test results for differences in the total *D. carinata* population sizes and absolute number of dispersers after 96 h of dispersal, depending on food availability in patch 1 and food availability in patch 2. Replicates with food available in patch 1 produced a significantly higher number of dispersers ($P < 0.01$) despite having significantly lower total population sizes ($P < 0.01$). Standard ANOVA assumptions were not violated.

**Table 2 Statistical results for proportion of individuals dispersing after 96 h of dispersal.**

| Parameter | Estimate (SE) | z stat | P-value |
| --- | --- | --- | --- |
| Intercept | −0.938 (0.100) | 9.371 | <0.001 |
| Food available in patch 1 | −2.872 (0.265) | 10.84 | <0.001 |
| Food available in patch 2 | −0.0206 (0.150) | 0.138 | 0.890 |
| Food available in patch 1 × Food available in patch 2 | 0.375 (0.350) | 1.073 | 0.283 |

Note:
Test results for differences in the proportion of *D. carinata* dispersing after 96 h depending on food availability in patch 1 and food availability in patch 2. A generalised linear model was used with parameter estimates on the logit scale and binomial errors. Food being available in patch 1 led a significantly lower proportion ($P < 0.001$) of individuals dispersing. Model variance was checked for overdispersion and did not violate standard GLM assumptions.

between patch 1 food availability and density ($t = 0.856$, $P = 0.404$; Table A1) despite the seemingly steeper increase in dispersal when patch 1 was not fed (Fig. A1).

An analysis based on absolute numbers in patch 2, rather than proportions, yielded the same overall trends in dispersal across our feeding treatments. Here, we found a significant difference in the total number of individuals in patch 2 according to whether patch 1 had been fed or not ($F_{1,16} = 13.605$, $P < 0.01$; Table 1), but no significant effect of food availability in patch 2 ($F_{1,16} = 0.102$, $P = 0.754$; Table 1). Indeed, patch 1 unfed groups had a far higher number of individuals in patch 2 despite their significantly lower total population sizes (individuals in patch 2: food-deprived patch 1, mean = 25.1, SE = 5.12; well-fed patch 1, mean = 4.3, SE = 1.73). Dispersers were also overwhelmingly juveniles (Fig. A2), although not disproportionately so ($z = 2.068$, $P = 0.331$; Table A2).

## DISCUSSION

In our system, there was a significant increase in inter-patch dispersal when *D. carinata* were deprived of food (Fig. 2), indicating that *D. carinata* exploited intra-patch information to inform their dispersal decisions. By contrast, extra-patch conditions

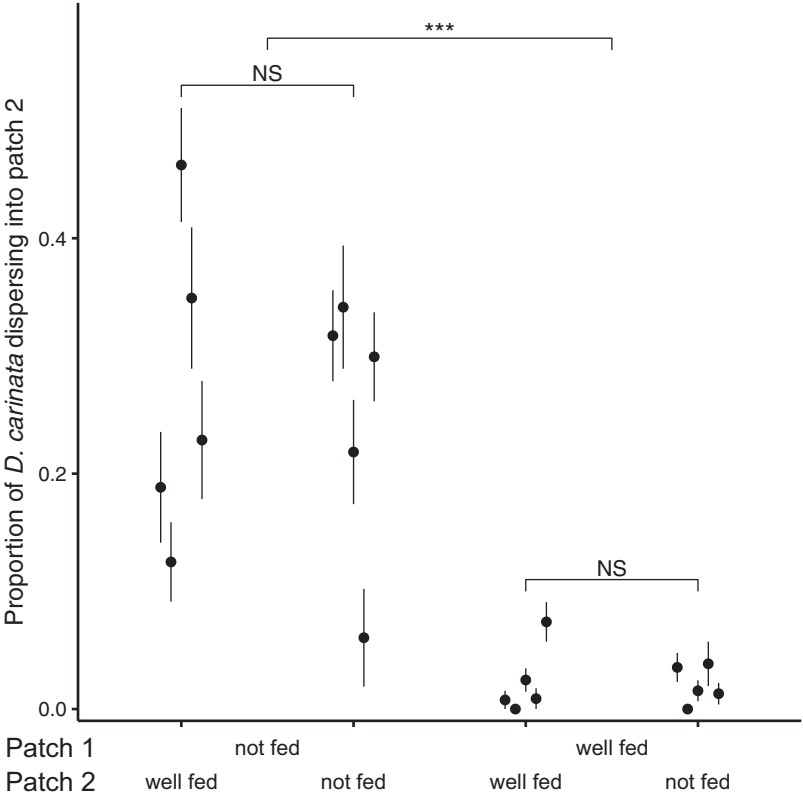

**Figure 2 Proportion of D. carinata dispersersing after 96 h across distinct feeding regimes.** The effect of 96 h of food deprivation on the proportion of *D. carinata* that had dispersed into patch 2, both with and without food available in patch 2 ($n = 5$ container pairs per treatment combination). ***indicates a significant difference between treatment combinations (at $P < 0.001$), NS indicates a non-significant difference. Each point and line is given by the mean number of *D. carinata* individuals in patch 2 as a proportion of the total population size between the two patches ± SE. SE was calculated as $SE = \sqrt{p(1-p)/n}$, where $p$ is the proportion of dispersers and $n$ is the number of individuals in the container pair. Food availability in patch 1 alone was found to have a significant effect on the proportion of the population that dispersed (Table 2).            

(food either abundant or entirely absent) had no effect on dispersal rates. Thus it appears that *D. carinata* either did not use, or were incapable of using, extra-patch information to inform their dispersal decisions.

Our first result—that animals increase dispersal propensity when faced with local resource shortages—has been well established empirically. Studies on taxa ranging from plants to invertebrates and vertebrates either imply, or experimentally demonstrate, that resource shortage is a powerful piece of information motivating dispersal (*Bowler & Benton, 2005*; *Martorell & Martínez-López, 2014*; *Fronhofer et al., 2018*). Our study adds *D. carinata* to the long list of organisms that exploit this piece of intra-patch information. Due to the seeming generality of this phenomenon across taxa (as established by *Fronhofer et al. (2018)*), it also appears likely that comparable results would be seen for other species of *Daphnia*, although additional studies using different clone lines or species of *Daphnia* may be prudent. Indeed, our results can only strictly be said to apply to the single clone line of *D. carinata* that we tested.

That the dispersal we observed was largely driven by resource shortage, rather than density, becomes obvious when examining patch 1 population sizes across treatments. Density, independent of resource shortage, has been demonstrated to cause changes in life-history in *Daphnia* spp. (*Matveev, 1993*; *Burns, 1995*, *2000*); but here, intra-patch resource shortage proved to be a far more powerful driver of dispersal than density in and of itself. Since the nature of our experimental design precluded any attempt to control density, those treatments that were well fed in patch 1 kept growing over time compared to those that were not, manifesting in a significantly higher total population sizes by the end of the experiment (Fig. 1). Despite these higher densities, which would not only have created more potential dispersers but also exacerbated any density-driven push effect, a significantly greater number of individuals dispersed in the treatments experiencing lower densities. Density did possibly drive some dispersal, with higher total population sizes increasing the proportion of the population dispersing regardless of whether patch 1 was fed or not (Fig. A1), although this effect was not found to be statistically significant (Table A1). Likewise, no significant effect was found for the interaction between density and food availability in patch 1, despite the apparent difference in the slopes of the patch 1 fed and unfed groups seen in Fig. A1. The quasibinomial error structure used in our GLM and the relatively low number of replicates means that any conclusions concerning the effects of density should only be drawn with caution; further investigation of the specific effects of density may be warranted.

The surprising result that food availability in patch 2 had no effect on final patch 2 population size (which should have been influenced by both births and deaths within the patch) was likely due a combination of factors. Since almost all dispersers were juveniles (Fig. A2), there was limited potential for births to occur in patch 2, regardless of food availability. Moreover, even if coming from a starved patch 1, those individuals that migrated to patch 2 had to be healthy enough to disperse in the first place, presumably depressing the death rate there. Nonetheless, had we recorded population sizes in patch 2 beyond 96 h, it seems likely that some difference would have soon become apparent between fed and unfed patch 2 populations.

The magnitude of the dispersal increase we observed also indicates that the effect of local resource information on dispersal rates may be pronounced. In terms of the proportion of dispersers, 25.9% of individuals dispersed into patch 2 under food deprivation, whereas less than a tenth of that (2.18%) did so under well-fed conditions (Fig. 2). Although this particular measure may have been inflated by the population growth that continued to occur in the well-fed treatments, the large difference in the absolute number of dispersers (food-deprived patch 1, 25.1 dispersers; well-fed patch 1, 4.3 dispersers) despite the afore-mentioned higher density in the well-fed treatments reiterated the strength of the effect. This suggests that ecological models may benefit substantially by accounting for conditional factors, like resource availability, that may have a large effect on dispersal behaviour.

We caution, however, that parameterising models using our or similar results should only be undertaken with great care. Such findings are likely to be influenced by the size and arrangement of the experimental set-up. Additionally, this study and comparable

efforts using *Daphnia* or other organisms typically rely upon highly unnatural environments within which to measure dispersal and movement (*Young & Getty, 1987*; *Dodson et al., 1997*; *Larsson & Kleiven, 1996*; *Roozen & Lürling, 2001*; *Fronhofer et al., 2018*), which may prompt aberrant behaviours. In our set-up in particular, it is possible that some individuals may not have recognised the dark tunnel opening as a dispersal avenue, or that individuals in general may have only encountered the opening infrequently. The length and width of the connections between our patches were likewise considerably smaller and narrower in scale than dispersal avenues might be expected to be in natural settings (*Michels et al., 2001*). Conducting equivalent experiments in natural environments and at scales relevant to the dispersal of the organisms under examination is likely to be useful, although admittedly difficult in practice.

Our second result—that favourability of conditions in the second patch had no effect on dispersal—highlights the apparent importance of push vs pull factors in driving a population's movement. In our study, to obtain information that would draw *D. carinata* into the second patch individuals either had to engage in prospecting within the inter-patch tunnel and the second patch, or to sense extra-patch conditions remotely. We found no evidence to suggest that *D. carinata* was capable of exploiting either source of information. In terms of more direct means of gathering information, extra-patch information gathering behaviours like prospecting are predicted to be costly due to the threat of predation that comes from moving into novel environments (*Bonnet, Naulleau & Shine, 1999*; *Hiddink, Kock & Wolff, 2002*; *Bonte et al., 2012*), or the simple energetic cost of having to move to assess new patches (*Delgado et al., 2014*). In *D. carinata* specifically, it seemed much more likely that chemoreception would serve as the primary means of ascertaining extra-patch conditions, as chemical signals from both conspecifics and other organisms have been demonstrated to have a multitude of effects on *Daphnia* growth and behaviour (*Larsson & Dodson, 1993*; *Dodson et al., 1994*). Indeed, it has been previously shown that *D. magna* and *D. pulex* are unaffected by olfactory cues from algae (*Roozen & Lürling, 2001*), but that a *D. galeata* and *D. hyalina* hybrid responds to them (*Van Gool & Ringelberg, 1996*). Here, however, the dominance of resource limitation in pushing dispersal from the local patch indicated that the pull to move into new patches was relatively weak in comparison, either because obtaining more information was costly, or because that information was in some way imperceptible or ignored.

## CONCLUSIONS

In conclusion, our results add to the growing body of evidence that condition-dependent dispersal is the norm amongst taxa, and that it is capable of generating substantial differences in dispersal behaviour (*Legrand et al., 2015*; *Fronhofer et al., 2018*). This growing empirical consensus warns against the simplifying assumption—used in the majority of ecological and evolutionary models—that dispersal rate is constant with respect to conditions. Relaxing that assumption is now well justified on empirical grounds, and the magnitude of shift in dispersal resulting from condition dependence suggests that it will have non-trivial effects when incorporated into mechanistic models of evolution, population dynamics, invasion spread, and so on. Amongst others, these effects may

include increased local adaptation within populations (*Armsworth, 2008*; *Armsworth & Roughgarden, 2008*), heightened risk of overcrowding (*Armsworth, 2008*), greater metacommunity stability (*Fronhofer et al., 2018*), and differences in metapopulation and invasion dynamics (*Neubert, Kot & Lewis, 2000*; *Kokko, 2006*; *Clobert et al., 2009*; *Schreiber & Lloyd-Smith, 2009*). In this light, the relative use of extra- vs intra-patch information is important because, when we move to a conditional dispersal model, the obvious simplifying assumption is that organisms exploit only intra-patch information. Our results suggest that intra-patch information is dominant in *D. carinata*, but the degree to which this is true generally will determine how complex our models of dispersal really need to be.

## ACKNOWLEDGEMENTS

We thank Katrina-Lee Ware and Hannah Edwards for their invaluable assistance in the preparation of the experimental set-up. We also thank Pepijn Luijckx, Christina Buesching, and Nicolas Schtickzelle for their comments on drafts of the manuscript.

### Funding
This work was supported by funding provided by the Australian Research Council (DP160101730; FT160100198). The funders had no role in study design, data collection and analysis, decision to publish, or preparation of the manuscript.

### Grant Disclosures
The following grant information was disclosed by the authors:
Australian Research Council: DP160101730; FT160100198.

### Competing Interests
The authors declare that they have no competing interests.

### Author Contributions
- Philip Erm conceived and designed the experiments, performed the experiments, analysed the data, prepared figures and/or tables, authored or reviewed drafts of the paper, approved the final draft.
- Matthew D. Hall analysed the data, authored or reviewed drafts of the paper, approved the final draft.
- Ben L. Phillips conceived and designed the experiments, analysed the data, authored or reviewed drafts of the paper, approved the final draft.

### Data Availability
All experimental data and scripts are available in the figshare repository: ERM, PHILIP (2019): CDD data and scripts. University of Melbourne. Fileset. https://doi.org/10.4225/49/5b0f62dc23b4c.

## Supplemental Information

Supplemental information for this article can be found online at http://dx.doi.org/10.7717/peerj.6599#supplemental-information.

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
