# Peer review of "Anywhere but here: local conditions motivate dispersal in Daphnia"

_PeerJ, doi:10.7717/peerj.6599_

## Round 0.1 · original submission · Major Revisions

Dear author,
Your paper has been reviewed by three experts in the field. All three reviewers see merit in the work, however all three also suggest that the paper requires Major Revisions before being acceptable for publication.

All three reviewers highlight that additional context of the work across species/groups need to be included, for example how these themes are addressed in mammal groups and other species. The reviewers quote specific papers (e.g. Proc B (reviewer 1), Current Zoology (reviewer 2), Nature Ecology and Evolution (reviewer 3)), which their inclusion may help contextualising this work.

Reviewers also highlight that the discussion could be expanded.

All reviewers are complementary about the study design, however reviewers one and two ask for clarifications that need to be addressed.
Reviewer three also raises some concerns around the validity of the results, which should be discussed or addressed.

There may also be opportunity for the figures to be improved (see reviewer suggestions).

We look forward to receiving your corrected paper for further evaluation.

Kind regards

Dr. Andrew Byrne

·

Basic reporting

The manuscript is well written, the structure conforms to PeerJ’s standard, figures and tables are clear and referred to in the text and reference list is complete. I do however have a few suggestions to improve the manuscript.

Line 22. (“resource availability in the alternative patch….” Instead of “resource limitation in the alternative patch…”
Line 23. Our work highlights the very large influence….. Limit the use of adverbs, they generally do not add anything to the findings.
Line 63. Still, broader Ecological and population dynamics may also be powerfully influenced by the relative…..
Line 105. Better use a reference to a published paper instead of an internet reference (e.g. modified as by Ebert 1998 using only 5% of the recommended SeO2 concentration”)
(reference to AdaM: D. Ebert, C. D. Zschokke-Rohringer, H. J. Carius, 1998. Within–and between–population variation for resistance of Daphnia magna to the bacterial endoparasite Pasteuria ramosa. ProcB 265: 2127–2134.)
Line 183. “Despite this advantage in”…. Replace with “Despite this higher….”
Line 190. ….may be extremely profound.
Line 197. …may have a very large effect….
Lines 263, 271, 293, 303, 320. Capitals mid-sentence in the references.
Line 307. Missing the “ : ” behind invited review.
Lines 233-347. Carefully check the references for Latin names and ensure they are in italic.
Tables. I assume the “¡” in the tables is used to denote significance? Potentially include this in the table legends. Or considering that in the figure *** are used be consistent and also use *** in the tables.
Figure 1. The authors could consider a slightly modified version of their figure were graph titles are on the outside (see attached file). This may make the figure slightly easier to read (although the current version is already good)
Figure 2 Similar comment as figure 1, the readability of the axis may be slightly improved (see attached file).

Experimental design

The experimental design is rigorous, has sufficient replication and addresses the main question. I just have a few small points that I would like to see clarified.

Could the authors clarify how the standard errors in figure 2 were obtained? I may have missed it but in the material and methods it states that the animals were counted every 24 hours. I suspect the authors did more than one count (and hence were able to provide SE for each replicate population). If this is indeed the case the material and methods should be amended.

As noted in the legend of table 1 standard anova assumptions were not violated. Could the authors amend the material and methods with the methods used to make these assessment and also provide test statistics on homogeneity of variance and normality?

Considering the authors did find clear differences between treatments I assume that the feeding differences between both patches are maintained even after the connection between both patches is opened. It may however help if the authors provided some additional information on how long difference between patches were maintained despite the direct connection between both patches. How long does it take for the algae to disperse (by diffusion) tough the entire system after being fed to patch 1?

Validity of the findings

On lines 179 to 188 the author’s discus the evidence for that dispersal was driven by resource shortage rather than density itself. To make their argument the authors compare the densities and number of migrants between the well fed and non-fed patch 1 populations. Although the points the authors make are valid the argument doesn’t take into account any interactive effects between density and resource shortage. Potentially density could still play a role in moderating strength of the response e.g. a stronger response at higher population densities. I wonder if the authors can strengthen their argument by correlating the number of migrants within each treatment to density (taking into account the higher baseline of migrants expect at higher densities)?

Additional comments

The authors provide a neat and well-designed experiment to understand how Daphnia use information from local and alternative patches to make a decision to stay or move. I found the paper quite interesting but believe it can be further improved by providing a deeper discussion.

I believe the a slight elaboration of the consequences and implications and the addition of a few concrete examples (e.g. in species X migration decisions are based both Y and Z) to support literature statements throughout the manuscript will help drive home the importance of these statements, the author’s findings and implications for the field and may make the paper appeal to a wider audience.

Building on my previous point I think that especially line 222-224 “Relaxing that assumption is now well justified on empirical grounds, and the magnitude of shift in dispersal resulting from condition dependence suggests that it will have non-trivial effects when incorporated into mechanistic models of evolution, population dynamics, invasion spread, and so on.” Could use additional explanation and examples. How may our theoretical predictions change, what actual effects?

·

Basic reporting

The manuscript is generally very well written and clear with a logical structure. The use and number of the included figures and tables is appropriate and serves to make the methods and results easily accessible. The figures are well designed and the layout of the tables is clear.
Nevertheless, some of the sentences are very long and could easily be divided into two (or more) by replacing a connecting “and” with a full stop (e.g., L 11: “…dynamics and it is apparent increasingly …”; L 30: “…, and the variety..”; L 53: “(…) and while they are…”). Please check the entire manuscript for such cases and change throughout.

Experimental design

This is a very interesting manuscript addressing an important paradigm. The research questions are well defined, and in the context of the current conservation landscape very timely.

I do, however, have some concerns/ questions regarding the experimental design: I am not a Daphnia specialist, and thus this may be a ‘stupid question’, but is there a possibility that within this genetically identical experimental population (L101), the genetic make-up governing dispersal decisions – and more specifically the (non-)gathering of extra-patch information – underwent mutation and is thus not representative for the entire species? Please address this in the text by either highlighting this possibility or by citing references on other ‘genetically inert behaviours’ substantiating that this is not possible.
Also, how likely is it for (habitat exploring) Daphnia (that are trying to gather extra-patch information) to find this single 15mm small dispersal tunnel? i.e., How robust are your conclusions that Daphnia does not use extra-patch information for dispersal decisions vs they would under natural circumstances, but could/ did not find the tunnel in your experimental design?
In addition, your tunnel length of 117mm seems quite a long distance for a small organism like Daphnia – for the reader to assess your experimental design, you would need to put this into context of known D. sp. dispersal distances under natural (or even different experimental) conditions!
In addition, two, sometimes fundamentally different, factors are important in shaping dispersal decisions: Somatic fitness and reproductive fitness. Somatic fitness is largely determined by food resources, as investigated in this manuscript. Reproductive fitness, however, is (also) affected by genetic components and thus population density/ number of available partners for sexual reproduction. Although the authors have got an experimental design (and the available data) to test these effects, they choose to largely ignore them. Nevertheless, even if sexual reproduction (and thus simple population density) is not that important in Daphnia sp. due to parthenogenesis, this aspect needs to be addressed at least in the introduction. Recent publications (e.g. Byrne et al. 2018) would help to introduce this paradigm.

Validity of the findings

Currently, this ms is focused purely on dispersal behaviour of Daphnia spp. limiting its appeal to a wider readership. However, with little additional work, this appeal could be broadened considerably by expanding the cited literature to a wider range of species (or even, simply naming the species in the text to which the currently cited references pertain). Specifically, throughout the introduction, I am missing examples on dispersal decisions governed/ moderated by intra- and/ or extra-patch resource availability from other species. In part, this could be solved by adding species names in the paragraph starting L40ff.

Additional comments

I very much enjoyed reading this manuscript. It sets out to investigate a timely paradigm addressing push/ pull factors affecting the dispersal behaviour of Daphnia spp., which could easily be expanded to other species.

General comments:
1) The manuscript is generally very well written and clear with a logical structure. The use and number of the included figures and tables is appropriate and serves to make the methods and results easily accessible. The figures are well designed and the layout of the tables is clear.
Nevertheless, some of the sentences are very long and could easily be divided into two (or more) by replacing a connecting “and” with a full stop (e.g., L 11: “…dynamics and it is apparent increasingly …”; L 30: “…, and the variety..”; L 53: “(…) and while they are…”). Please check the entire manuscript for such cases and change throughout.
2) Throughout the introduction, I am missing examples on dispersal decisions governed/ moderated by intra- and/ or extra-patch resource availability from other species. In part, this could be solved by adding species names in the paragraph starting L40ff.
3) I am not a Daphnia specialist, and thus this may be a ‘stupid question’, but is there a possibility that within this genetically identical experimental population (L101), the genetic make-up governing dispersal decisions – and more specifically the (non-)gathering of extra-patch information – underwent mutation and is thus not representative for the entire species? Please address this in the text by either highlighting this possibility or by citing references on other ‘genetically inert behaviours’ substantiating that this is not possible.
Also, how likely is it for (habitat exploring) Daphnia (that are trying to gather extra-patch information) to find this single 15mm small dispersal tunnel? i.e., How robust are your conclusions that Daphnia does not use extra-patch information for dispersal decisions vs they would under natural circumstances, but could/ did not find the tunnel in your experimental design?
In addition, your tunnel length of 117mm seems quite a long distance for a small organism like Daphnia – for the reader to assess your experimental design, you would need to put this into context of known D. sp. dispersal distances under natural (or even different experimental) conditions!
4) Two, sometimes fundamentally different, factors are important in shaping dispersal decisions: Somatic fitness and reproductive fitness. Somatic fitness is largely determined by food resources, as investigated in this manuscript. Reproductive fitness, however, is (also) affected by genetic components and thus population density/ number of available partners for sexual reproduction. Although the authors have got an experimental design (and the available data) to test these effects, they choose to largely ignore them. Nevertheless, even if sexual reproduction (and thus simple population density) is not that important in Daphnia sp. due to parthenogenesis, this aspect needs to be addressed at least in the introduction. Recent publications (e.g. Byrne et al. 2018) would help to introduce this paradigm.
Specific comments:
L85: rephrase: “a greater range of responses has been observed” (i.e., “range” is singular)
L113ff: rephrase for ease of understanding: “…data collection. Each container measured 90mm x 75mm at the base, was 110mm high and widened gradually towards the top to 100mm x 90mm. A circular hole…”
L118: Again, as a non-Daphnia expert, how long would it take an individual to traverse this 117mm dispersal tunnel? This is important to add here so that readers can assess your experimental design of feeding every 24h and, particularly, counting population sizes in both patches after 96h (L140).
L122: rephrase: “… we examined the effects of …”
L125: You are giving an average of 122 individuals/ patch here; surely this has to be an exact measure to assess effects of resource availability/ abundance and social factors accurately? At the bare minimum, you need to include min & max as well as STD here!
L148: Again, I realise that for non-Daphnia specialists it would be helpful if you’d include average lifespan of species somewhere above, so that all readers can assess the results of unfed populations shrinking over the course of 95h (L147ff & Fig. 1)
L153: Shouldn’t this read “during the first 96h” instead of “at 96h”?
L163ff: Interesting that food availability affects population size in patch 1, but not in patch 2. I assume, this must be because population density was always below carrying capacity in patch 2? But then, there were unfed patch 2’s, too, where I would expect Daphnia to starve and die even if they did get ‘pushed’ to disperse into this unfavourable habitat through lot food in natal patch 1. And the reverse is true, too: In favourable patch 1’s, population increased – why did this not happen in patch 2 (i.e., independent of immigration, but simply through higher breeding and survival)? Maybe my non-understanding is partly again due to my lack of knowledge of Daphnia life expectancy and generation span - Please clarify!
L175: Please add species to the cited references in this paragraph!
L178: You infer that “due to the generality of dispersal being influenced by intra-patch information, your results can be extra-polated to other D. spp, too – However, to assess this inference, the reader needs to be informed if there are species where intra-patch resource availability has been shown to NOT affect dispersal decisions!
L179ff: This result would be better included as a testable hypothesis and prediction, rather than simply being added here in the discussion! This would strengthen the appeal of this ms to a wider readership considerably! As it stands, you are currently presenting your study simply as investigating resource availability push/ pull factors, but neglecting to formally investigate also population-density dependent PUSH factors – although you do have the appropriate experimental design and data!
L206ff: How likely/ possible is it for chemical information to make its way through this narrow and fairly long dispersal tunnel? I assume there ius no current/ water movement at all between the 2 tanks. This again highlights my concern that the findings of this study cannot easily be transferred to natural environments and conclusions drawn here have to be assessed with care when making inferences to natural D. sp populations!

·

Basic reporting

This article is well written. It states the research question clearly and places it within the current literature. Extra references to current research on the impact of resources on dispersal could be cited, e.g.
Fronhofer E.A., Legrand D., Altermatt F., Ansart A., Blanchet S., Bonte D., Chaine A.S., Dahirel M., De Laender F., De Raedt J., di Gesu L., Jacob S., Kaltz O., Laurent E., Little C.J., Madec L., Manzi F., Masier S., Pellerin F., Pennekamp F., Schtickzelle N., Therry L., Vong A., Winandy L. & Cote J. (2018) Bottom-up and top-down control of dispersal across major organismal groups. Nature Ecology and Evolution, 2, 1859–1863.

Figures could be improved. In particular, figure 2 seems to present very similar data as figure 1 (see below). Figure 1 should not be formatted as a histogram (to be reserved for count data) but rather as a mean +- confidence interval (or other measure of error/variation among the 5 replicates, which is currently missing).

Some improvements to the tables are possible too. E.g. in table 1 statistics relate to count at 96h but this is not specified. The caption tile of table 1 is also very uninformative. It seems "<" has been incorrectly typed/converted in tables 1 and 2.

Experimental design

Methods are briefly but quite clearly described. Some details should be added to make them completely clear and allow readers to understand what has been done and potentially redo the experiments:
* Daphnia individuals were counted (L.127) but it should be stated how, and especially if the census was exhaustive (all individuals were counted) or based on a sample.
* A justification should be given to the length of the connecting pipe (corridor) between the two habitat patches. This is linked to my comment about the need for a better justification about the use of extra-patch information (see below).

Validity of the findings

My main criticism about the validity of the findings is related to the statement that “local conditions alone drive dispersal”, already present in the title. I believe this point is overstated for several reasons. First, the absence of an effect cannot be proved, by nature. Of course, in practice one can reach a reasonably high level of certainty about the absence of an effect provided the power to detect it, if it exists, is high enough. In the present study, I find the introduction lacks some discussion about why it could be expected that Daphnia could acquire some information about the conditions in the distant patch, together with an associated justification of the experimental setup, in particular the length of the connecting corridor pipe. Indeed, individuals cannot ignore something they do not perceive. Stating that this information is ignored implies to first prove that this information can be known, e.g. because individuals have been observed to travel back and forth and “sample” the distant patch, or can sense some chemical molecules from a distance. Some pieces of information related to that are given in the discussion (L. 199 ff).

If such information about perception cannot be given, I believe that the conclusion of the absence (or ignorance) of the extra-patch information by the individuals to base their dispersal decisions is speculative, and should be explicitly stated that way.

Also, but less importantly, some discussion about the potential differences in the results if other Daphnia clones had been tested would be nice, given the introduction states (L. 88) that impacts on dispersal can vary among Daphnia species and clones.

---

## Round 0.2 · Minor Revisions

Dear authors,

Your paper has again been reviewed by three experts in the field, and the consensus is that the paper has been improved significantly, however minor revisions are still required before the paper is acceptable for publication.

Reviewer 1 suggests that one additional assessment is made of the comparison of slopes, to help with increasing the robustness of your inference. Reviewer 3 also suggests that appropriate edits to the figures provided be made, outside of R programming environment, if necessary.
More importantly, reviewer three suggests that the title of the paper be toned down, and I would suggest you edit the title with regards to the premise the "local conditions alone" drive dispersal within this system.

We look forward to receiving your corrected paper.

Kind regards

Dr. Andrew Byrne

·

Basic reporting

Line 50. Replace or remove “powerfully”?
Line 72. “low density patches” instead of “vulnerable” patches”?
Line 81. ….will help to resolve questions of such character, maybe better, will help to resolve these/such questions.
Line 82. “Here, we examine their relative………” Considering it is the first sentence of the paragraph maybe use; “Here, we examine the relative influence of intra-patch and extra patch information on dispersal by……”
Line 176. Could the authors also provide the P-value for the relationship?
Line 185. Taking into account the age distribution (i.e. number of juveniles) in the origin patch were there more migrating juveniles than expected by chance?
Line 201. I would write, “Indeed, strictly our results only apply to the single clonal line of D. carinata we tested.
Line 219. Exchange “no practical potential” for “limited potential”, and remove “there”…..e.g. “..was limited potential for births to occur in patch 2, regardless of food availability.
Line 234. We caution, however, that parameterising models using our or similar results should only be undertaken with great care. The results are likely to depend………………………
Line 247. Maybe replace “in the present case” with “In our study”?
Line 264. I would remove the “moreover” from the sentence. “In conclusion, our results add to the growing body of evidence that condition-dependent dispersal is the norm amongst taxa, and that it is capable of generating substantial differences in dispersal behaviour (Legrand et al. 2015; Fronhofer et al. 2018).”

Experimental design

No comments, the authors have addressed all my concerns satisfactory.

Validity of the findings

I would like to thank the authors for adding the additional supplementary figures. However as noted already on Line 176 for figure A1 besides the amount of variation explained by the correlation (r2) I would also suggest testing if the correlation itself is significant and if the slopes between both lines differ. This would allow you to provide more certainty in the sections starting at line 243

Additional comments

The authors have done a good job at addressing the issues raised by the reviewers. The additional examples listed in introduction and discussion and the more in-depth discussion have greatly improved this manuscript. Besides the few issues I have raised above I have no further concerns and would recommend a minor revision.

·

Basic reporting

No comment

Experimental design

no comment

Validity of the findings

no comment

Additional comments

I am impressed by the thoroughness with which the authors addressed reviewer concerns on the previous draft. All my queries have been implemented (or rebutted) to my satisfaction and I very much like the revised version of this manuscript.

Two very small points:
- In line 24, please change "large influence" to "considerable ..." and
- In line 248 please change "The seemingly odd result" to "This surprising result"

·

Basic reporting

The authors improved their manuscript based on the three reviews they received. In particular, precisions about some literature cited were added in the introduction.
Figures were improved but to some extent. In particular, if profoundly disagree with the following justification: “Figure 1 has been altered in accordance with the recommendation insofar as our plotting software allows us to do so (ggplot2 for R). Figure 2 has been left as is, since using the approach that would be necessary with ggplot2 (faceting, as in figure 1) would unfortunately prevent us from labelling which groups are significantly different in a clear manner.” I found this to be a very good example of the “tyranny of the tool”. R is a great tool, but it is just a tool, and many other tools still exist. If your tool cannot do what you want, why not using another tool?! As an example, you could add extra items (such as labels) on the figure plotted with ggplot2, simply by using an image software or even Powerpoint.

Experimental design

The authors added key information about the design, which make the experimental design, and its limitations, clearer. However, some statements in the rebuttal are still more “personal beliefs” than “proven reality”. For example, the fact that “Daphnia carinata are highly active swimmers and most individuals repeatedly encounter the opening each day” does not seem supported by data. I agree with the conclusion that given the mobility of Daphnia individuals, it is likely that they encounter the corridor entrance; but to state “they repeatedly encounter the opening”, I’d like to see some data.

Validity of the findings

My main criticism about the validity of the findings was related to the statement that “local conditions alone drive dispersal”, already present in the title. I believe this point is overstated. Although the authors introduced more nuanced statement at several places in the manuscript, I still believe the title should be changed. Indeed, the title is likely to be understood by readers as some kind of proof that other sources of information than local information are ignored: “local conditions alone drive dispersal” does not mean “local conditions is the only factor we were able to show to have some effect on dispersal”. But the manuscript does not prove that at all. If the experiment was able to show that some information is perceived but not used, I would agree. But this is not the case. Given the importance of the title in conveying the general message of a scientific study, I strongly advise to modify it to state only what is proven in the study: Daphnia take local conditions into account in their dispersal decision, so dispersal is not random. I understand this may be less sexy because there are already a series of studies showing this result, but this is the only conclusion supported by the data.

---

## Author Rebuttal · Round 0.2

# Rebuttal letter for "Anywhere but here: local conditions alone drive dispersal in *Daphnia*"

We would like to thank the reviewers for their thoughtful feedback and comments. We believe their suggestions have improved the manuscript immeasurably. For clarity, each of our responses is listed beneath a characterisation of the comment, rather than a direct quote from the review. Our responses were based on the reviews themselves however, rather than the characterisations given here.

**Reviewer 1**:
- Line by line suggestions.
  - All adopted with no exceptions.
- Table typographical error.
  - Corrected.
- Figures could be formatted more clearly according to the following guidelines.
  - Figure 1 has been altered in accordance with the recommendation insofar as our plotting software allows us to do so (ggplot2 for R). Figure 2 has been left as is, since using the approach that would be necessary with ggplot2 (faceting, as in figure 1) would unfortunately prevent us from labelling which groups are significantly different in a clear manner.
- How were the standard errors in figure 2 derived? And did you count once or multiple times?
  - The standard errors shown on the figure are for each container replicate's proportion of dispersers. They were calculated according to a formula that is typical for calculating the standard error of a binomial proportion. This formula has now been added to the caption of figure 2. Counts were performed once per day exhaustively by eye, and this how now been specified in the methods.
- Can you present some statistics that make the ANOVA assumption check explicit?
  - We have now amended the methods to state how we checked normality and homogeneity of variances, and reported the test statistics associated with our checks.
- Do patch conditions in linked containers equalise over time?
  - Although we didn't collect any specific data on how much algae was entering the inter-patch tunnel nor the other patch, it was clear from observation that algae fed into one patch did not visibly reach the other patch. Our algae species is non-motile (now noted in the methods) and always settled between feedings, so we judged that differences in patch condition were maintained between linked patches on different feeding regimes. This is also likely to be the case because there was no flow of liquid between linked patches that could have helped to transfer algae.
- Could an interaction between density and resource shortage have been affecting your results?
  - As reviewer 1 remarks, since we didn't control for density, it remains possible that there could have also been a separate effect of density, as well as an interaction between density and our resource availability treatments. To more closely examine this, we have now produced a supplementary figure that shows density versus the proportion of dispersers based on whether patch 1 was fed or not. We have expanded our density discussion paragraph to

interpret this figure; it appears that density may have had an effect, although it was small in comparison with resource availability.

- Some elaboration of consequences and implications might help.
    - To expand upon the consequences and implications of our research, we have now added a discussion paragraph that raises some cautionary points concerning the use of our results and those from similar experiments to parameterise dispersal models. Also see "relaxing the assumption" response below.
- Some examples when making literature statements might help.
    - See response to a similar comment by reviewer 2 below.
- Can you discuss the implications of "relaxing the assumption that dispersal is unchanging" in more detail?
    - We have elaborated on the two specific examples we cited in our concluding paragraph to demonstrate how moving from the assumption of fixed to conditionally-dependent dispersal may alter predicted evolutionary/ecological outcomes, as well as introduced its implications for metacommunity stability. We have also drawn attention back to some of the potential outcomes that were mentioned in our introduction.

**Reviewer 2:**
- There are a number of sentences that are too long. "And" is overused.
    - We've divided up or shortened the particular sentences pointed out by the reviewer. We've also shortened a number of other sentences.
- Is it possible that this is not representative of the entire species?
    - Yes. We now specifically highlight that fact in the discussion.
- How likely is it that the *Daphnia* weren't aware of the tunnel opening?
    - In our estimation unlikely, since a large proportion of individuals were able to find and use the tunnel in particular treatments. *Daphnia carinata* are highly active swimmers and most individuals repeatedly encounter the opening each day. Still, as the reviewer suggests, it is possible that individuals who would have otherwise dispersed did not because they couldn't find the opening, which may have depressed overall dispersal rates. A line acknowledging this idea has been added to the discussion paragraph on the experimental set-up/environment.
- How was the tunnel length selected? Is it long or short for *Daphnia*?
    - The equipment set-up that we use was primarily designed for undertaking experimental invasions with *Daphnia* as a model organism. As such, the lengths between patches were set to accord with the above goal, rather than with recourse to specific questions about *Daphnia* dispersal distances in natural settings. Admittedly this may limit the applicability of our findings, which we now highlight in the discussion paragraph on experimental set-ups. We don't consider the tunnel's length to be long for *Daphnia*, since an individual could easily traverse such a distance in less than a minute if travelling in a straight line. We now include a reference in support of the above statement for readers that may not be familiar with the speeds at which *Daphnia* can move.
- What of somatic fitness and reproductive fitness?
    - Unfortunately we have been unable to locate the reference ("Byrne et al. 2018") suggested by the reviewer, nor other references concerned with the distinction between somatic and reproductive fitness in an ecological context.

Since we see our manuscript as being concerned with the extent to which organisms use local and non-local information on resources to inform dispersal decisions, we feel that considering other factors that motivate dispersal is largely outside of our remit. However, it may be that we have missed the reviewer's point here as we haven't seen the reference.

- Can you draw out species and examples in the introduction?
    - In the introduction, we now relate the species that feature in a number of our citations and provide some examples. We also now highlight the variety of taxa that were shown by Fronhofer et al. 2018 to use resource availability and predator presence to inform their dispersal decisions.
- Line by line suggestions.
    - All accepted, with the following exceptions and notable cases:
        - L125. We've opted to remove the specific number of animals that was given here, as the purpose of the line was rather to explain that our populations contained individuals of a variety of age/sizes when dispersal began, and not to provide specific results that are already contained within figure 1.
        - L153. Most of our discussed results are of our counts that were taken 96 hours after dispersal commenced. As such, we have changed "at" to "after" rather than followed the specific suggestion of the reviewer.
        - L163ff. We believe that food availability in patch 2 did not have much of an effect on growth/die-off in patch 2 for a number of reasons that we now include in our discussion. Briefly stated here, they are: (1) individuals reaching patch 2 were overwhelming juveniles (shown in the newly added Figure A2), which were incapable of reproducing whether patch 2 was provisioned or not; (2) individuals reaching patch 2 had to be healthy enough to successfully disperse, and so were less likely to die-off even when patch 2 was not provisioned; (3) dispersers had only been in patch 2 for between 1—4 days, so detectable differences in die-off and population growth in patch 2 may not have had time to manifest yet.
        - L175: With the exception of Martorell and Martínez-López 2014 (the focal species of which is now communicated in our introduction), we have deliberately avoided listing specific species from either Bowler and Benton 2005 or Fronhofer et al. 2018. This is because both papers deal with many species, as the former is a review with 5 species in the food availability section, and the latter a distributed "metaexperiment" that examined 21 species. Thus, rather than exhaustively list all 26 species, we instead opted for the general statement that "Studies on taxa ranging from plants to invertebrates and vertebrates either imply, or experimentally demonstrate, that resource shortage is a powerful piece of information motivating dispersal", which we believe summarises the most salient aspect of the references. As highlighted below, we also now expand upon our discussion of Fronhofer et al. 2018 in the introduction to make sure that readers have a better impression of the reference if they are not familiar with it.
        - L178: As mentioned above, in our introduction we now expand upon the landmark study by Fronhofer et al. (2018) in which it was shown that local resource levels were a powerful motivator of dispersal in all 21 species that the authors examined, which spanned taxa as diverse as

protists, slugs, crustaceans, crickets, newts, and fish (amongst others). We now also refer back to this study more obviously in our discussion to support our assumption that it's likely that similar behaviours would be seen in other species of *Daphnia*.

- L179ff: As discussed in relation to a similar query about the effects of density on our results by reviewer 1, we have now extended our results and discussion to include the role that density may have played in shaping the dispersal we observed. We now communicate this potential role in our abstract. Since we did not specifically control for density in our experimental design however, we believe that our experiment only makes a minor contribution to the question of density and dispersal, and so don't believe that it is necessary to represent it as a major aim of the experiment in our introduction.
- L206ff: As stated in response to reviewer 1, there was no substantial flow between our linked patches. *Daphnia* occupy still water environments, and so this particular aspect of the experiment accords with their natural habitat. However, as detailed in our discussion of the tunnel length above, we now have a paragraph in the discussion that acknowledges the potential limitations of extrapolating our results to natural environments.

**Reviewer 3:**

- You should cite (or perhaps cite more) Fronhofer et al. 2018.
  - We now cite the published version of this reference, rather than the preprint. The published version, released just last month, had not come out when we were first drafting the manuscript. We also take additional time to explain the scope and significance of the paper in the introduction.
- Figure 2 seems to present very similar data to figure 1.
  - Since we dealt with the proportion and absolute counts of dispersers separately, we felt that having one figure for each was necessary. Figure 2 also has the additional advantage of displaying the variation seen both across and within (from the standpoint of a binomial proportion) our container pair replicates.
- Figure 1 should not be formatted as a histogram (to be reserved as count data) but rather as a mean +/- CI (or other measure of variation, which is currently missing).
  - We believe that there may be some confusion as regards the difference between a bar chart and a histogram here. Our figure is a stacked bar chart (used for depicting counts/frequencies across categorical variables), rather than a histogram (used for depicting the distribution of numerical data across a continuous variable). We have, however, now added error bars to show variance.
- Table results do not specify that they relate to counts at 96 hours.
  - Corrected.
- Table typographical error.
  - Corrected.
- The caption title of table 1 is uninformative.
  - Since the tables are intended to do no more than present statistical test results, we are not sure how to give them more informative titles than they presently have. The captions have been expanded to highlight the significant results however.

- How did you count specifically?
  - We now state in the methods how we conducted our counts (once per day exhaustively by eye).
- What's your justification for the length of the connection?
  - Repeated from our answer to reviewer 2:
    "The equipment set-up that we use was primarily designed for undertaking experimental invasions with *Daphnia* as a model organism. As such, the lengths between patches were set to accord with the above goal, rather than with recourse to specific questions about *Daphnia* dispersal distances in natural settings. Admittedly this may limit the applicability of our findings, which we now highlight in the discussion paragraph on experimental set-ups. We don't consider the tunnel's length to be long for *Daphnia*, since an individual could easily traverse such a distance in less than a minute if travelling in a straight line. We now include a reference in support of the above statement for readers that may not be familiar with the speeds at which *Daphnia* can move."
- It seems unlikely that *Daphnia* could sense extra-patch conditions. Saying that they might ignore it is not strong enough, as it implies that it is knowable. You should be clear that the absence/ignorance of extra patch information is speculative. Or else use less definite language.
  - It is a fair point that it is possible (and indeed likely) that *Daphnia* were simply incapable of detecting extra-patch conditions (rather than just ignoring extra-patch information). Emphasising that idea was the intention of our original statements towards the start of the discussion that "it appears that *D. carinata* either did not use, *or were incapable of using*, extra-patch information" and towards the end of the discussion that "[patch 2 information possibly had no effect] because that information *was in some way imperceptible* or ignored", however we recognise that the language used elsewhere in the manuscript may have been too definite. We have now amended sections of the abstract and discussion to better reflect that it is likely that *Daphnia* are simply incapable of detecting extra-patch conditions. We believe that the title does not overstate our findings since local conditions were indeed found to be alone in affecting dispersal rates in our experiment, presumably due to the likely imperceptible nature of extra patch conditions (which, as stated, we now give additional emphasis in the rest of the manuscript).
- Some discussion about potential differences in the results if other *Daphnia* clones had been tested would be nice, since it's known that there's clonal variation.
  - We have now added a statement at the end of the second discussion paragraph that our results can only strictly be said to apply to a single clone line of a single species. However, we believe that our now expanded introduction and discussion (in particular as regards Fronhofer et al. 2018) helps to more clearly justify our assumption that the phenomenon is likely to be general across *Daphnia* spp.

Thank you,

Philip Erm, Matthew D. Hall, and Ben L. Phillips

---

## Round 0.3 · accepted · Accept

Dear authors,
Having reviewed your updated manuscript, I am satisfied that you have addressed all of the three reviewers suggestions satisfactorily.

Congratulations on a very interesting paper.

Kind regards

Dr. Andrew Byrne

#